# Maternal Vitamin D Levels during Pregnancy and Offspring Psychiatric Outcomes: A Systematic Review

**DOI:** 10.3390/ijms24010063

**Published:** 2022-12-21

**Authors:** Subina Upadhyaya, Tiia Ståhlberg, Sanju Silwal, Bianca Arrhenius, Andre Sourander

**Affiliations:** 1Research Centre for Child Psychiatry, INVEST Flagship, University of Turku, 20014 Turku, Finland; 2Public Health Stations, City of Helsinki, PB 6310, 00099 Helsinki, Finland; 3Department of Child Psychiatry, Turku University Hospital, 20521 Turku, Finland

**Keywords:** maternal, prenatal, vitamin D, offspring, psychiatric outcomes, 25(OH)D

## Abstract

Prenatal exposure to vitamin D may play a significant role in human brain development and function. Previous epidemiological studies investigating the associations between maternal vitamin D status and offspring developmental and psychiatric outcomes in humans have been inconclusive. We aimed to systematically assess the results of previously published studies that examined the associations between maternal vitamin D levels, measured as circulating 25(OH)D levels in pregnancy or at birth, and offspring neuropsychiatric and psychiatric outcomes. Systematic searches were conducted using MEDLINE, Embase, PsychINFO and Web of Science for studies published by 10 August 2022. We included human observational studies that examined associations between prenatal or perinatal vitamin D levels and offspring neuropsychiatric and psychiatric outcomes and were published in English in peer-reviewed journals. Of the 3729 studies identified, 66 studies were screened for full texts and 29 studies published between 2003 and 2022 were included in the final review. There was a small amount of evidence for the association between prenatal vitamin D deficiency and autism spectrum disorder. When studies with larger sample sizes and stricter definitions of vitamin D deficiency were considered, positive associations were also found for attention-deficit/hyperactivity disorder and schizophrenia. Future studies with larger sample sizes, longer follow-up periods and prenatal vitamin D assessed at multiple time points are needed.

## 1. Introduction

Vitamin D is essential for bone mineralization and bone mass acquisition [1]. It may also be important for the development of other organ systems including the central nervous system [2]. In the past few decades, the role that vitamin D plays in brain development and function has been explored. Abundant vitamin D receptors have been found in brain regions such as the prefrontal cortex and hippocampus. Vitamin D is important for neuronal differentiation and reducing apoptosis in the hippocampus, the area involved in language and memory [3]. It serves as a transcriptional regulator by expressing genes vital to brain development [4,5].

Vitamin D is crucial especially when the central nervous system of the fetus develops and the brain is sensitive to maternal nutritional deficiencies [6,7]. Previous literature has emphasized the role of early prenatal insults in relation to later mental illnesses. The Dutch famine study was one of the earliest epidemiological studies of this kind and it suggested associations between maternal nutritional deficiencies during pregnancy and the subsequent risk of offspring mental illness. Offspring who were exposed to prenatal nutritional deficiency had a twofold risk of schizophrenia later in life, compared to the unexposed cohort [8].

Furthermore, studies from rodents have suggested morphological changes in the brain due to prenatal exposure to vitamin D deprivation in utero [9]. Animal studies suggest that vitamin D plays an important role in fetal brain development. However, human studies that have investigated the associations between maternal vitamin D status and offspring developmental and psychiatric outcomes in humans have been inconclusive. Previous systematic reviews and meta-analyses have explored associations between prenatal vitamin D and some neurodevelopmental or cognitive outcomes including autism spectrum disorder (ASD) and attention-deficit/hyperactivity disorder (ADHD) [4,10]. The reviews reported contradictory results. The systematic review found inconclusive evidence for offspring neurocognitive and psychological outcomes [4] whereas the meta-analysis reported potentially decreased risks of ADHD and autism-related traits with increased exposure to prenatal vitamin D [10]. The first study only included studies published up to 2014, while the other study did not include any other psychiatric outcomes than ASD and ADHD.

Research interest in the possible prenatal effects of vitamin D has been growing and many studies published in the last five years have emphasized the need to summarize and review the rapidly gathered new information. That is why the main aim of this review was to systematically assess the results of previously published studies that examined the relationship between maternal vitamin D levels, measured as circulating 25(OH)D levels in pregnancy or at birth, and offspring psychiatric outcomes. Previous studies focused on neurodevelopmental disorders and that is why we wanted to broaden the study area to encompass other psychiatric outcomes, including both disorders and symptoms.

## 2. Materials and Methods

This systematic review was conducted in accordance with the Preferred Reporting Items of Systematic Reviews and Meta-analyses (PRISMA) [11] and the Synthesis Without Meta-analysis reporting guideline [12]. The review protocol was prospectively registered with The Open Science Framework (osf.io/jt4wa) [13].

### 2.1. Search Strategy

Relevant studies were identified by comprehensive searches of electronic databases from inception until 10 August 2022. We conducted systematic searches in MEDLINE, Embase, PsychINFO and Web of Science by combining terms that covered maternal vitamin D during pregnancy or at birth, the use of serum samples, and offspring psychiatric outcomes. Potentially relevant papers were also searched using the backward snowballing technique, which involves looking at the reference lists of the included papers and screening them to find other possible papers to include. The search strategy was developed and refined after consultation with a library information specialist (Appendix A).

### 2.2. Inclusion and Exclusion Criteria

The review included human observational studies that used longitudinal study designs and were published in peer-review journals. The included papers were written in English and measured associations between circulating 25(OH)D levels during pregnancy or from newborns and psychiatric outcomes in offspring. As maternal and neonatal vitamin D levels have been shown to be correlated, we also included studies that measured vitamin D levels in cord blood or neonatal dried blood samples [14]. The outcomes in the included studies reported offspring neuropsychiatric or psychiatric outcomes based on registry diagnoses, diagnostic interviews or symptoms.

We excluded any cross-sectional studies, as well as case reports, conference abstracts, editorials, comments, letters and reviews with no relevant primary data.

### 2.3. Study Selection Procedures

The studies were retrieved from search databases and checked for duplicates. Two reviewers (B.A./T.S., S.W./S.U.) screened the titles and abstracts independently, followed by full-text assessments of the papers based on the inclusion and exclusion criteria. Two reviewers (B.A./T.S., S.W./S.U.) cross-checked the papers and any disagreements were resolved after discussion.

### 2.4. Quality Assessment

The quality assessment of the studies was conducted by two independent researchers (B.A./T.S., S.W./S.U.) using the Joanna Briggs Institute (JBI) critical appraisal tools specific to study designs [15]. The reviewers used the JBI tool with 10 questions for case-control studies and the tool with 11 questions for cohort studies to assess the methodological quality of the studies. The tools enable the studies to be categorized as yes, no, unclear or not applicable. Any discrepancies with the JBI scores were resolved through discussion and studies were only included if both reviewers agreed.

### 2.5. Data Extraction and Synthesis

After the quality assessment, the relevant data were extracted and placed in an Excel spreadsheet. The data included the author, year, country, study design, sample size, offspring age at assessment, vitamin D assessment time, vitamin D categorization, outcome assessment, limitations and associations between vitamin D and outcome. We summarized the most relevant information in two tables: the first covered psychiatric disorders or diagnoses and the other was for the outcomes of psychiatric symptoms.

## 3. Results

### 3.1. Study Selection

The electronic search using the MEDLINE, Embase, PsychINFO and Web of Science databases yielded 3729 titles. After we removed any duplicates, the number of publications included in this review was 2444. Further screening of the titles and abstracts yielded 66 papers for the full-text review and 28 were found to be relevant. One additional article was found in the references of the included papers. The flow diagram for the screening processes and study selection are shown in Figure 1.

### 3.2. Characteristics of the Included Studies

The 29 included studies were published from 2003 to 2022 (and 16 of them were published during the last five years of the search period (2017 to August 2022). The studies were from four regions: three from Australia [16,17,18], three from China [19,20,21], 17 from Europe (for specific countries see Table 1a,b) [22,23,24,25,26,27,28,29,30,31,32,33,34,35,36,37,38] and six from the USA [39,40,41,42,43,44]. There were 14 cohort studies, 8 had a nested case-control design, either within a cohort or trial, and 7 were case-control studies. The covariates included in the studies varied greatly. Most studies included prenatal, perinatal and socio-demographic factors, but maternal psychopathology was addressed in 10 studies [19,21,24,27,29,30,32,33,36,38].

The outcomes for this review were limited to neuropsychiatric or psychiatric disorders and symptoms and therefore neurocognitive outcomes were not reported. The outcomes in the studies varied, but a clear majority of the studies had examined ASD and ADHD (Figure 2). Some studies had examined diagnoses, which were retrieved from clinical registers or were determined by a clinician during the study. Other studies examined symptoms measured by questionnaires. Some of the questionnaire scores were analyzed by using dichotomous yes/no categorizations for certain cut-off points and others were studied as continuous variables. Most of the studies examined outcomes in children and adolescents (Figure 2). Few studies had followed the subjects into young adulthood [16,24,27,31] and only one study examined adult outcomes [40]. The age of the offspring when the outcomes were assessed in each study are presented in detail in Appendix A. The vitamin D values were measured at different time points, varying from early pregnancy to neonatal samples (Figure 3). Only four studies had measured vitamin D concentrations at more than one time point pre- and perinatally [34,35,38,44]. Vitamin D was studied as a continuous variable (22 studies), by clinical categories (18), deciles (2), tertiles (1), quartiles (6) and quintiles (4) (Figure 4). Vitamin D levels were presented as nmol/L in 22 studies or as ng/mL in 7 studies. The definitions for deficient levels varied greatly. In the studies that used nmol/L, the deficiency was defined as between <20 nmol/L and <50 nmol/L. The study characteristics have been described in detail in Table 1a,b. The findings of the reviewed studies have been summarized in detail by each outcome, in Table 2.

### 3.3. Quality Assessment

The included studies had adequate quality, as assessed by the JBI quality appraisal tools. The biggest concerns in the reviewed studies were attrition rates, insufficient strategies used to address attrition, scarce confounders, small sample sizes and insufficient length of follow up. Vitamin D levels were often measured from a subsample and it was unclear if this was because of attrition or based on a planned decision. The sample sizes for the studied outcomes varied greatly and the number of cases with diagnosed outcomes was between 24 and 1558. In addition, the number of cases whose mothers were vitamin D deficient was relatively low in some studies. The follow-up time was considered insufficient in some of the studies, because the offspring’s age at assessment was less than five years. Diagnoses based on self-report questionnaires were excluded from the review but symptom-level outcomes of those studies were included [16,28,33]. Furthermore, one study assessed both ASD symptoms and diagnoses, but there were only three diagnosed cases and the ASD diagnoses outcomes were not included [18]. The quality appraisal is presented in detail in Appendix A.

### 3.4. Vitamin D and Offspring Neuropsychiatric and Psychiatric Outcomes Reported by Each Outcome

#### 3.4.1. ADHD and ADHD Symptoms

ADHD was studied as a diagnostic outcome in four studies [26,31,32,44] and ADHD symptoms in five studies [19,22,28,29,37]. There were 24 [31] to 1067 [32] cases with diagnosed ADHD. Diagnoses of ADHD were either made by a clinician [26], register-based data [31,32] or parental reports [44]. The ADHD symptoms were reported by parents in all but one study, where teacher reports were used [28]. We did not include the diagnostic outcomes from that study, as we did not consider that teacher reports on their own were a valid way of diagnosing ADHD.

Two studies reported significant links between maternal vitamin D levels and the increased odds of offspring being diagnosed with ADHD [32,44]. The largest study that examined ADHD outcomes reported significant odds ratios for both continuous and categorized vitamin D levels and ADHD [32]. The other study with significant associations reported an association between low third trimester values and increased odds for offspring ADHD but no association for values measured in early pregnancy [44].

The other two studies of diagnosed ADHD did not find significant associations between ADHD and continuous or categorized vitamin D levels from umbilical cord samples [26,31] or categorized levels from maternal sera from the third trimester [31]. It is noteworthy that the second study defined vitamin D deficiency below <50 nmol/L, which could be considered a relatively high value compared to some other studies [31].

Three out of five studies on symptom-level outcomes [22,28,29] presented associations between low prenatal or perinatal vitamin D and an increased risk for ADHD symptoms, while the other two studies did not find clear associations [19,37]. Three of the studies measured vitamin D from maternal sera in early pregnancy [22,28,37] and two from cord blood [19,29]. Four studies only assessed toddlers or preschool-aged children [19,22,28,29], which might have had an impact on the validity of the results. One study found no difference in ADHD symptoms by vitamin D category but found an interaction related to vitamin D deficiency in the association between maternal depression and offspring ADHD symptoms [19].

#### 3.4.2. ASD and ASD Symptoms or Traits

Ten studies examined the associations between maternal vitamin D during pregnancy and offspring ASD diagnoses [20,21,23,25,30,34,38,41,42,43], whereas four studies examined ASD symptoms or traits [18,35,37,39]. The sample of ASD cases ranged from 58 [25] to 1558 [30]. The diagnoses were either made by a clinician [20,25], were register-based [30,38] or were based on medical records [34,42,43]. There was one study that included parental reports of ASD diagnoses [41].

Three studies showed significant associations between maternal vitamin D levels during pregnancy and ASD diagnoses in offspring [20,30,34]. They comprised two studies that examined maternal vitamin D in early pregnancy [20,30] and one study that examined mid-gestation samples [34]. The largest study, with 1558 ASD cases, showed significant odds ratios for the continuous, lowest quintiles compared to the highest quintiles and categorical vitamin D measures [30]. One study reported significant associations between maternal first trimester vitamin D levels in the lower three quartiles (quartiles 1, 2, 3), compared to the highest quartile (quartile 4) [20]. Another study reported an association between deficient vitamin D levels < 25 nmol/L in mid-gestation and offspring ASD, but no association for vitamin D measured from cord blood [34].

In contrast, five studies did not find any associations between maternal vitamin D levels during pregnancy and offspring ASD [23,38,41,42,43]. Although one study did not find any association in the overall sample, there was a significant association between maternal vitamin D insufficiency (25–<50 nmol/L) and ASD in Nordic-born mothers [38]. Most studies did not include race/ethnicity as a covariate. Two studies reported significant associations between neonatal vitamin D levels and ASD diagnoses in offspring [21,38] and one found similar associations but only in females [41].

ASD symptoms or traits in children were reported by parents in three studies [18,35,37]. Two studies showed significant associations between vitamin D levels and ASD symptoms or traits. One study reported that sufficient vitamin D levels during pregnancy (≥30 ng/mL) were associated with a lower number of ASD symptoms than deficient levels [37]. Another study reported significant associations between lower vitamin D concentrations <25 nmol/L in mid-gestation, or at the time of birth, and ASD traits [35]. However, one study did not find any correlation between low vitamin D during pregnancy and ASD traits [18].

#### 3.4.3. Depressive Disorders and Depressive Symptoms

No associations were found in the two studies that examined associations between vitamin D levels during pregnancy and diagnosed depression [31] or depressive symptoms [36]. The first study measured maternal vitamin D levels at 30 weeks of gestation and the depression diagnoses were obtained from registers. The subjects were either diagnosed or they had been given at least one prescription for antidepressant medication [31]. In the second study, vitamin D levels were measured at any time during pregnancy and depression symptoms were measured using the Short Mood and Feelings Questionnaire [36].

#### 3.4.4. Eating Disorder Symptoms

Eating disorder symptoms were only included in one study [16]. This was a cohort study and the offspring were assessed at 14, 17 or 20 years of age. Eating disorder symptoms or diagnoses were based on the Child Eating Disorder Examination or an Eating Disorder Examination Questionnaire. In addition, the subjects’ body mass indexes were measured and the females were asked about their menstruation. The study used both eating disorder and eating disorder symptoms as definitions for the outcomes and this meant that the diagnostic validity remained somewhat unclear. A total of 98 adolescents filled in the diagnostic criteria and only 13 were males. There were no associations in the total sample, but low maternal vitamin D levels were associated with eating disorders in female offspring. Parental psychopathology was not considered as a covariate.

#### 3.4.5. Schizophrenia, Schizoaffective Disorder and Psychotic Symptoms

Psychotic disorders were included in three studies [24,27,40] and one study reported psychotic symptoms and disorders based on a self-report questionnaire [33]. Only the symptoms from that study were included in this review. All three studies examined schizophrenia [24,27,40], and one study also included schizoaffective disorder [40]. The number of cases varied between 26 [40] and 1301 [24] and the offspring were followed until at least young adulthood in all three studies. The two register-based studies found an increased risk for schizophrenia [24,27] in subjects with neonatal vitamin D deficiency. Both studied neonatal dried blood samples and vitamin D levels as quintiles and continuous variables. Interestingly, one study found that the highest quintile had an increased risk for schizophrenia when the fourth quintile was used as a reference [27], while another study did not find any associations between maternal vitamin D levels in the third trimester and schizophrenia or schizoaffective disorder [40].

One study measured vitamin D from maternal sera at any stage of pregnancy and examined the association with psychotic experiences among 18-year-olds. The number of individuals with psychotic experiences was 117 out of more than 2000 subjects. No associations were established [33].

#### 3.4.6. Behavioral or Emotional Symptoms

Four studies assessed behavioral or emotional symptoms [17,22,37,39]. All four studies measured vitamin D levels from maternal sera in the first or second trimester. Two studies assessed very young children [22,39], whereas the other two had multiple assessment points in childhood and adolescence [17,37]. The Infant Toddler Social Emotional Assessment Scale [39], Child Behaviour Checklist (CBCL) [17,37] and Strengths and Difficulties Questionnaire (SDQ) [22,37] were used to assess behavioral and emotional symptoms.

One study reported fewer behavioral and externalizing symptoms in children whose mothers had higher vitamin D levels [22]. One study found an association between low maternal vitamin D levels and increased internalizing scores, but among children of White ethnicity [39]. Two studies did not find associations between prenatal vitamin D levels and offspring’s behavioral or emotional symptoms [17,37].

Since the studies were heterogeneous, we also examined these studies in several subgroups based on three factors: first, large sample size and stricter definitions of vitamin D deficiency and with at least 100 mothers in the deficient group (Table 3); second, time of measurement of vitamin D levels (Appendix A); and third, geographical locations. There was a tendency to report positive associations between prenatal vitamin D levels and offspring ASD, ADHD or schizophrenia for subgroups with large sample sizes and stricter definitions of vitamin D deficiency. However, there were no clear tendencies to report associations based on categorization by time of measurement of vitamin D levels. Moreover, we observed more tendencies to report positive associations from countries from Nordic regions.

## 4. Discussion

The present systematic review found inconclusive findings with regards to the associations between maternal vitamin D deficiency and neuropsychiatric and psychiatric outcomes. Compared to other outcomes, the most evidence was accumulated for the association between maternal vitamin D deficiency and ASD. When studies with larger sample sizes, stricter definitions for vitamin D deficiency and at least 100 mothers in the deficient group were considered, positive associations were also found for ADHD and schizophrenia. Moreover, studies from Nordic regions with low sunlight exposure during the winter season were more likely to report positive associations between vitamin D levels during pregnancy and offspring psychiatric outcomes. Only one study examined depressive disorder, depressive symptoms, eating disorder symptoms and psychotic experiences and no consensus could be reached for these outcomes. None of the studies examined anxiety or bipolar disorders.

### 4.1. Neuropsychiatric and Psychiatric Outcomes

The findings of this review suggest a small amount of evidence for any association between maternal vitamin D deficiency and offspring ASD diagnoses. This finding was in line with a previous systematic review that suggested the potential role of vitamin D during pregnancy and autism in children [45]. Possible mechanisms for this association are related to the role of vitamin D in brain development. For example, vitamin D seems to participate in the regulation of certain neurotransmitters, including serotonin [46] and dopamine [47]. Previous studies have shown that these neurotransmitters may function abnormally in subjects with ASD [48,49].

Interestingly, when we examined a subsample of studies with larger sample sizes, stricter definitions of vitamin D deficiency and at least 100 mothers in the deficient group, there was a tendency towards reporting more positive findings for ASD, ADHD and schizophrenia (Table 3). The evidence was inconclusive for symptom level outcomes, including ASD symptoms, ADHD symptoms, and behavioral and emotional symptoms. This was in line with previous reviews that examined neurocognitive outcomes and found mixed results [4,50,51]. The lack of associations in these studies may be attributable to factors such as small sample sizes, low prevalence of vitamin D deficiency, age at the time of the assessment and assessment tools. Many studies were based on small sample sizes that, for example, weakened the statistical power of possible subgroup analyses. While most of the studies controlled for maternal and neonatal factors, only eight studies adjusted for maternal psychiatric illness [19,21,24,27,29,30,32,40], and six for race or ethnicity [18,30,32,38,41,44]. These are both important confounding factors for the associations between maternal vitamin D levels and offspring psychiatric outcomes. One study only reported the association between maternal vitamin D levels and offspring behavioral and emotional symptoms by race, leaving out relevant information about the total sample [39]. In addition, when we examined studies by geographical locations, Nordic countries had a greater tendency to report positive associations. Moreover, the majority of these studies adjusted for the seasons when subjects were born or had blood drawn.

Several of the included studies had some important limitations. For example, the assessments were made at an early age in many studies, including studies on ADHD symptoms [19,22,28,29] and behavioral and emotional symptoms [39]. Furthermore, most of the questionnaire-based studies just relied on information from parental reports. In addition, attrition rates were high in most of the cohort studies and they were not always clearly reported. The included confounders varied: most studies had considered socio-demographic and pre- or perinatal risk factors, but parental psychopathology was included only in ten studies. Parental psychopathology is an important potential confounder because genetic effects are well established in the etiology of most neuropsychiatric and psychiatric disorders.

### 4.2. Timing of Exposure

The time when vitamin D was measured in the included studies varied from early pregnancy to neonatal samples. While most of the studies measured vitamin D levels in early pregnancy or dried blood spots from newborns, few studies had measured vitamin D levels during mid-gestation, late pregnancy or from umbilical cord blood at birth. The studies that measured vitamin D deficiency during early pregnancy or used newborn samples had a greater tendency to report positive associations. Only four studies had measured vitamin D status during multiple stages of pregnancy [34,35,38,44], which limited the ability to compare results across pregnancy. One study examined the association with ADHD using maternal vitamin D during the first and third trimesters and only found a significant association for vitamin D deficiency in the third trimester [44]. However, another measured vitamin D levels in mid-gestation and from a neonatal blood sample and observed significant associations with autistic traits at both time points [35]. Similarly, one study found significant associations between ASD and vitamin D levels in early pregnancy and the neonatal period, but the findings were restricted to the children of Nordic mothers [38]. In contrast, one study only found significant associations between ASD and maternal vitamin D levels in mid-gestation, but not for ASD and newborn vitamin D concentrations [34].

Previous literature suggests that maternal vitamin D concentrations can be seen as the determinant or the proxy of the closely correlated fetal vitamin D status. It also suggests that neonatal vitamin D concentrations measured from cord blood at birth can be used as markers of fetal vitamin D status at the end of gestation [14]. It is difficult to extract evidence from just the results of on these few studies measuring vitamin D levels at two time points in pregnancy. Moreover, these particular studies were based on small sample sizes. Future studies with large sample sizes and vitamin D measurements at multiple time points are warranted.

### 4.3. Categorization of Exposure

There was high heterogeneity in how vitamin D levels were classified in different studies. While 22 of the studies reported vitamin D levels using continuous variables, 18 reported varying clinical categories. The definitions of vitamin D deficiency varied considerably across studies, from <20 nmol/L to <50 nmol/, and this made it difficult to compare the findings. Moreover, there were few cases with deficient vitamin D levels in some studies, which could have limited the power to detect associations.

### 4.4. Age at Assessment

Most studies were conducted among young children, with a few among young adults or adults. Most associations were observed in young children, especially for ASD and ADHD. Only one study followed up on offspring behavioral symptoms until young adulthood and found no associations in the adolescent samples [17]. Based on two studies on schizophrenia, there were some indications that possible adverse effects of vitamin D deficiency might persist into young adulthood [27,40]. However, more studies are needed to confirm these findings. In some studies, the age range of the children was provided, and the studies were categorized by mean age at assessment for comparison; thus, some misclassification may have occurred.

### 4.5. Strengths and Limitations

A key strength of this systematic review is that we focused on a broad range of neuropsychiatric and psychiatric outcomes. This scope has not been explored by previous reviews that have focused on vitamin D in relation to neurodevelopmental or neurocognitive outcomes [4,10]. Furthermore, we investigated the studies rigorously by looking at the timing of vitamin D measurements, the categorization of the vitamin D levels and the age when psychiatric outcomes were assessed. However, there are several limitations that should be considered. First, several previous reviews focused on cognitive, motor and language development outcomes and prenatal vitamin D and we did not include those outcomes in our review. Second, the studies included in the review were heterogeneous. The study populations varied, especially by the timing of the exposure measurements and the categorization of vitamin D, which made it difficult to harmonize the results across studies. Third, only a few studies investigated outcomes other than ASD or ADHD, which prevented us from drawing firm conclusions on most of the other psychiatric outcomes. For example, eating disorders, psychotic experiences, depressive disorders and depressive symptoms were only examined in one study each. None of the studies examined other common psychiatric outcomes, such as anxiety and bipolar disorders. Fourth, only four studies measured vitamin D levels at two time points and the results were not uniform across the different time points. Finally, there were few studies from low-to-middle-income countries and this means that the most of the findings were based on high-income countries.

## 5. Conclusions

Our systematic review provides a small amount of evidence for the association between maternal vitamin D levels during pregnancy and offspring ASD. When studies with larger sample sizes and stricter definitions of vitamin D deficiency were considered, positive associations were also found for ADHD and schizophrenia. The findings could have important implications for public health, as vitamin D deficiency can be readily prevented with vitamin supplements. However, as the included studies were all observational, no conclusions can be made regarding the causality of the observed associations. Several other factors including genetic factors also play a role in the multifactorial etiology of neuropsychiatric and psychiatric disorders. No consensus was established for other psychiatric outcomes due to inconclusive findings or the low number of studies. Further studies with larger sample sizes and longitudinal designs with long follow-up periods are needed to draw firm conclusions, especially for adulthood-onset disorders. Moreover, future studies should preferably measure vitamin D at multiple time points during pregnancy, and at birth, to establish a window that might be sensitive to vitamin D deficiency. In particular, more studies are needed to explore the associations between prenatal vitamin D and some currently understudied common psychiatric outcomes such as anxiety, depression and bipolar disorders.

## Figures and Tables

**Figure 1 ijms-24-00063-f001:**
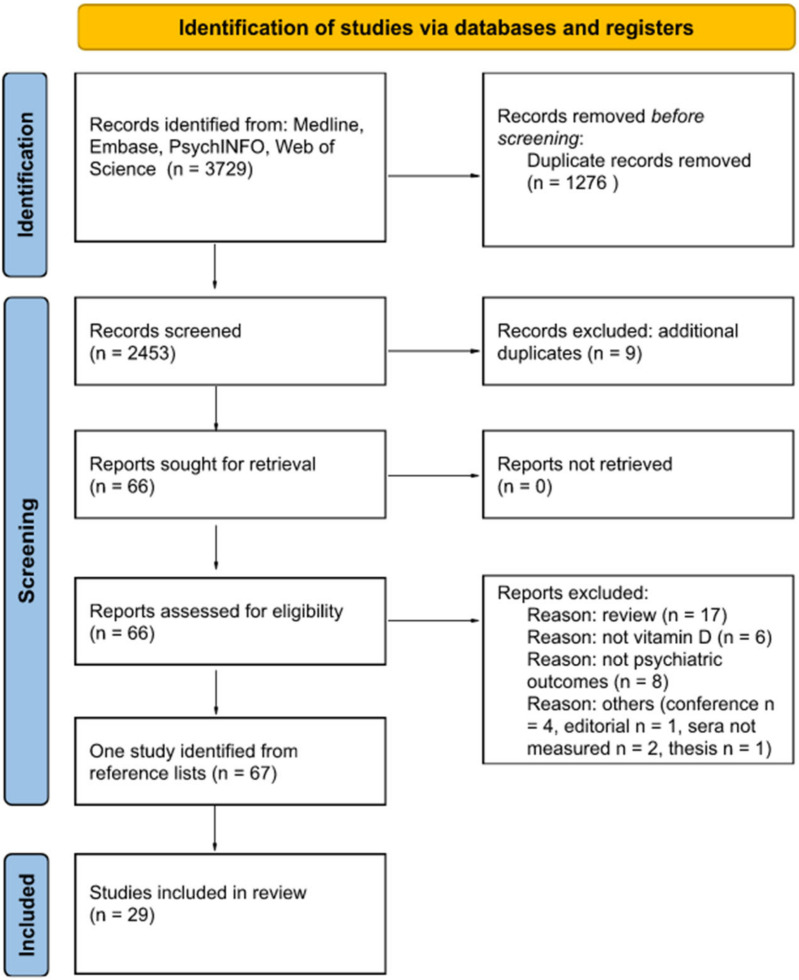
PRISMA flow diagram of the study.

**Figure 2 ijms-24-00063-f002:**
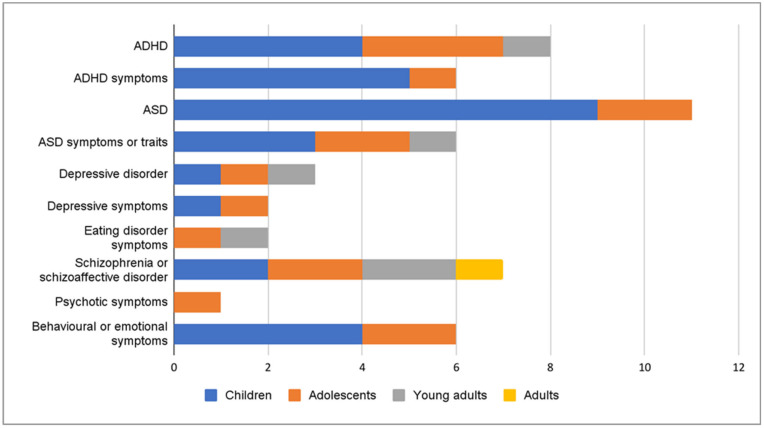
The number of studies for each outcome, stratified by the age groups assessed in the study. Some studies examined multiple outcomes and multiple age groups.

**Figure 3 ijms-24-00063-f003:**
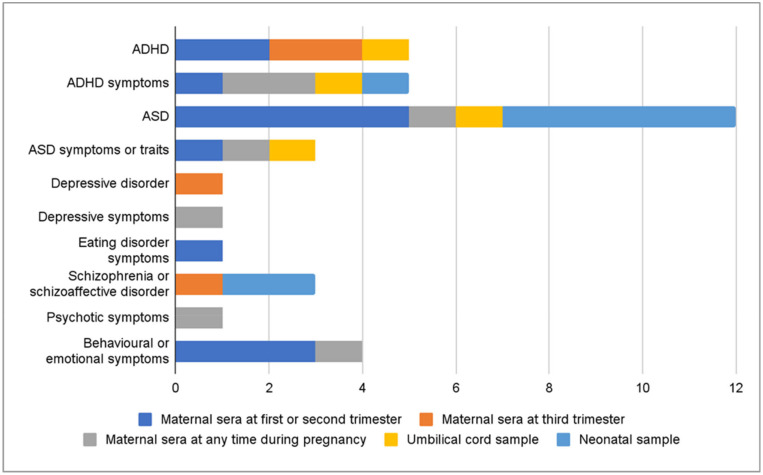
The number of studies for each outcome, stratified by vitamin D assessment time points in the study. Four studies assessed vitamin D at multiple time points.

**Figure 4 ijms-24-00063-f004:**
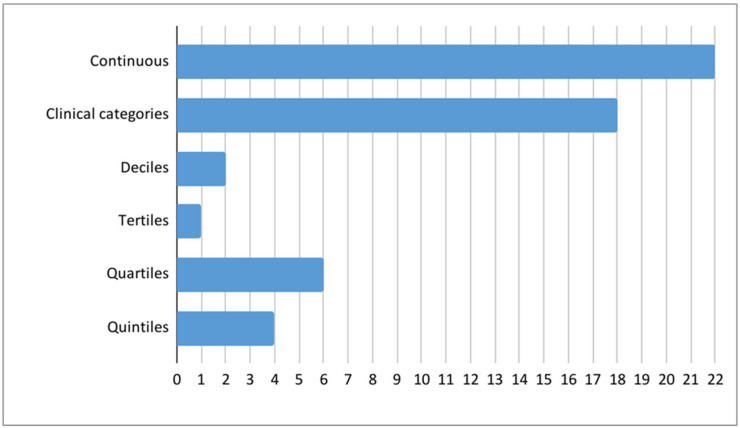
The number of studies by the methods of examining vitamin D levels. Many studies had more than one way to examine vitamin D levels.

**Table 1 ijms-24-00063-t001:** (**a**) Summary of papers with psychiatric disorders or diagnoses as the outcome. (**b**) Summary of papers with psychiatric symptoms or questionnaire scores as the outcome.

**(a)**									
**Author, Year, Country**	**Study** **Design**	**Psychiatric** **Disorders in Offspring**	**N: Total Cohort/Outcome** **Disorder; or Cases/Controls**	**Age at** **Outcome** **Assessment**	**Vitamin D** **Assessment Time**	**Vitamin D** **Categorization**	**Outcome Assessment**	**Limitations**	**Main Findings**
Attention-deficit/hyperactivity disorder (ADHD)									
Chu et al., 2022 [44] USA	Nested sample from a randomized study	ADHD	879/124	6–9 years	Maternal sera at 10–18 weeks and 32–38 weeks of gestation	Categories: <12 ng/mL, 12–19.9 ng/mL, 20–29.9 ng/mL and ≥30 ng/mL	Parental report of clinical diagnosis of ADHD		No association between vitamin D at 10–18 weeks and ADHD. At 32–38 weeks: deficient aOR 0.34 (95% CI 0.12–0.94) and insufficient aOR 0.41 (95% CI 0.15–1.10) categories against the highly deficient vitamin D category.
Gustafsson et al., 2015 [26] Sweden	Case-control	ADHD	202 cases, 202 controls	5–17 years	Umbilical cord blood samples	Continuous, quartiles, 10th and 25th percentiles	Clinical diagnoses	Controls not matched by sex; limited covariates considered	No associations: continuous vitamin D and ADHD: OR 0.99 (95% CI 0.96–1.01), levels below the 25th percentile: OR 1.13 (95% CI 0.69–1.83), levels below the 10th percentile: OR 1.14 (95% CI 0.56–2.34).
Sucksdorff et al., 2021 [32] Finland	Nested case-control	ADHD	1067 cases, 1067 controls	2–13.7 years; median age 7.3 years (SD 1.9)	Maternal sera at first or early second trimester	Continuous, categories: <30 nmol/L, 30–50 nmol/L, >50 nmol/L and quintiles	Register-based diagnoses *ICD-10*: F90.0, F90.1, F90.8 or F90.9	Limited coverage of all ADHD children from registers	Significant associations: continuous decreasing vitamin D levels and offspring ADHD: aOR 1.45 (95% CI 1.15–1.81), lowest versus highest quintile aOR 1.53 (95% CI 1.11–2.12).
Strøm et al., 2014 [31] Denmark	Cohort	ADHD Depression	965/24 ADHD cases, 102 depression cases	ADHD: median age 17.2 years. Depression: median age 19.3 years	Maternal sera at 30 weeks of gestation	Continuous and categories: <50 nmol/L, 50–75 nmol/L, ≥75–125 nmol/L, and ≥125 nmol/L.	Register-based diagnosis of ADHD: *ICD-8*: 308, *ICD-10*: F90 or psychostimulant prescription. Depression: *ICD-8*: 2960, 2962, 2968, 2969, 2980, 3004 or 3011, *ICD-10*: F32) or antidepressant prescription.	Small sample of ADHD cases	No significant associations between vitamin D levels <50 nmol/L and ADHD or depression. Higher vitamin D concentrations showed a direct association with offspring depression (*p* trend = 0.02)
Autism spectrum disorders (ASDs)									
Egorova et al., 2020 [23] Sweden	Case-control	ASD	100 cases, 100 controls	N/A	Maternal sera at 14 weeks of gestation	Continuous	Register-based diagnoses *ICD-10*: F84.0 or F84.5	Study focused on several biomarkers: no specifics available for vitamin D	No association between continuous vitamin D and ASD: OR 0.77 (95% CI 0.57–1.04).
Chen et al., 2016 [20] China	Case-control	ASD	68 cases, 68 controls	3–7 years; mean 3.9 years (SD 1.2)	Maternal sera at 11–12 weeks of gestation	Continuous and quartiles	Diagnosis confirmed by clinician using *DSM-V* criteria	Small sample size	Rising continuous vitamin D was associated with lower odds of ASD: aOR 0.862 (95% CI 0.795–0.913). Quartile analyses: first quartile: aOR 3.99 (95% CI 2.58–7.12), second quartile: aOR 2.68 (95% CI 1.44–4.29) compared to the fourth quartile of vitamin D concentration.
Fernell et al., 2015 [25] Sweden	Nested case-sibling	ASD	58 cases, 58 siblings	>4 years	Dried blood spots from newborns	Continuous	Clinical diagnoses	Small sample size, no covariates considered	The mean vitamin D level was lower in children with ASD (24.0 nmol/L, SD 19.6) than in their siblings (31.9 nmol/L, SD 27.7), and the difference was significant (t57 = 2.57, *p* = 0.013, d = 0.33).
Lee et al., 2021 [38] Sweden	Nested case-control, case-sibling	ASD	Pregnancy sample: 449 cases, 574 controls; neonatal sample: 1399/1607; neonatal sibling sample: 357/364	0–17 years	Maternal sera at 10–11 weeks of gestation and dried blood spots from newborns	Categories: <25 nmol/L, 25–50 nmol/L and ≥50 nmol/L	Register-based diagnoses *ICD-9*: 299, *ICD-* *10*: F84, and *DSM-IV* 299	Stratified analyses limited due to sample size	Maternal vitamin D levels were not associated with ASD in the overall sample. In the subgroup of Nordic-born mothers, there were associations between maternal vitamin D insufficiency (25–50 nmol/L) and ASD: OR 1.58 (95% CI 1.00–2.49). Neonatal vitamin D < 25 nmol/L was also associated with ASD: OR 1.33 (95% CI 1.02–1.75) compared to levels ≥ 50 nmol/L.
Schmidt et al., 2019 [41] USA	Case-control	ASD	357 cases, 234 controls	Mean age at diagnosis: 3.6 years (SD 0.8)	Blood samples from newborns	Continuous and categories: <50 nmol/L, 50–<75 nmol/L, ≥75 nmol/L	Service use from regional medical centers. ASD diagnosis confirmed by parental ADI–R interview and child assessment with ADOS.	Few subjects had vitamin D deficiency	No association between a 25 nmol/L increase in vitamin D levels and ASD before or after adjustment OR 0.96 (95% CI 0.86–1.06) and aOR 0.97 (95% CI 0.87-1.08). For females only (N = 47) there was an association between higher vitamin D and less ASD diagnoses: OR 0.74 (95% CI 0.55–0.99).
Sourander et al., 2021 [30] Finland	Nested case-control	ASD	1558 cases, 1558 controls	0–18 years; median age at diagnosis: 6.6 years (SD 3.2)	Maternal sera at first or early second trimester	Continuous, categories: <30 nmol/L, 30–50 nmol/L, >50nmol/L and quintiles	Register-based diagnoses *ICD-10*: F84.x and *ICD-9*: 299.x		Increasing continuous vitamin D levels and decreasing risk of offspring ASD: aOR 0.75 (95% CI 0.62–0.92). Lowest quintile aOR 1.36 (95% CI 1.03–1.79) compared with the highest quintile. Deficient category (<30 nmol/L) aOR 1.44 (95% CI 1.15–1.81), and insufficient (30–49.9 nmol/L) aOR 1.26 (95% CI 1.04–1.52), compared to sufficient levels.
Vinkhuyzen et al., 2017 [34] the Netherlands	Cohort	ASD	9778/68 cases	~6 years	Maternal sera in mid-gestation and cord blood at birth	Continuous and categories: <25 nmol/L, 25–49.9 nmol/L and ≥50 nmol/L	Screening with parent-reported questionnaires (SRS, CBCL, SDQ), medical records and parental interviews of screen-positive children. Polygenic risk scores included.	Small sample of ASD cases	Maternal serum at mid-gestation: deficient versus sufficient OR 2.42 (95% CI 1.09–5.07). Cord blood: deficient versus sufficient, no association.
Windham et al., 2019 [42] USA	Case-control	ASD	562 cases, 426 controls	4–9 years	Dried blood spots from newborns	Continuous and categories: <50, 50–75 and >75 nmol/L	Medical records, expert review, final case status confirmed if *DSM* criteria were met	Few newborns were deficient (14% had vitamin D < 50 nmol/L)	No association between vitamin D levels and ASD, some significant interactions with ethnicity and sex.
Windham et al., 2020 [43] USA	Case-control	ASD	534 cases, 421 controls	N/A	Maternal sera at mid-gestation	Continuous and categories: <50, 50–75 and >75 nmol/L	Medical records, expert review, final case status if *DSM* criteria were met	Few mothers were deficient (<10%)	No overall association between vitamin D and ASD or protective effect with increasing levels of vitamin D, but some significant interactions with ethnicity and sex.
Wu et al., 2018 [21] China	Nested case-control	ASD	310 cases, 1240 controls	3 years	Dried blood spots from newborns	Continuous, quartiles, deciles and over or under 30 nmol/L	Clinical examination (*DSM-V* diagnostic criteria) ADI-R, parent report	Possible overadjustment with covariates	The median vitamin D level was lower in children with ASD compared to controls (*p* < 0.0001). RR for the lowest first quartile: 3.6 (95% CI 1.8–7.2), second quartile: RR 2.5 (95% CI 1.4–3.5) and third quartile: RR 1.9 (95% CI 1.1–3.3) compared to the fourth quartile.
Psychotic disorders									
Eyles et al., 2018 [24] Denmark	Nested case-control	Schizophrenia	1301 cases, 1301 controls	5–24 years	Dried blood spots from newborns	Continuous and quintiles	Register-based diagnoses (*ICD-10*: F20). Polygenic risk scores accounted for in a subsample.	Youngest cases only 5 years at the time of diagnosis	The lowest quintile had an increased risk: IRR 1.44 (95% CI 1.12–1.85) compared to the highest quintile. In the combined sample with polygenic risk scores: lowest quintile IRR 1.52 (95% CI 1.20–1.93) compared to the highest quintile. Continuous vitamin D and ASD: IRR 0.92 (95% CI 0.86–0.99).
McGrath et al., 2003 [40] USA	Case-control	Schizophrenia, schizoaffective disorder	26 cases, 51 controls	Adults, age range not specified	Maternal serum at third trimester	Continuous	Expert review of interview data and medical records (using *DSM-IV* criteria)	Small sample size	There was no significant difference in the vitamin D levels of cases and controls OR 0.98 (95% CI 0.92–1.05). However, the vitamin D levels of subjects with black ethnicity differed somewhat (not statistically significant).
McGrath et al., 2010 [27] Denmark	Nested case-control	Schizophrenia	424 cases, 424 controls	11–24 years	Dried blood spots from newborns	Continuous and quintiles	Register-based diagnoses (*ICD-10* F20)	The results do not suggest a dose–response relationship between low vitamin D and schizophrenia.	Compared to the fourth quintile, the lowest quintile had an RR of 2.1 (95% CI 1.3–3.5), while those in the second and third quintiles had RRs of 2.0 (95% CI 1.3–3.2) and 2.1 (95% CI 1.3–3.4), respectively. The fifth (highest) quintile also had an increased RR of 1.71 (95% CI 1.04–2.8).
**(b)**									
**Author, Year**	**Study Design**	**Psychiatric Symptoms in Offspring**	**N for Total Cohort/Outcome** **Response**	**Offspring’s Age at** **Assessment**	**Vitamin D Assessment Time**	**Vitamin D** **Categorization**	**Outcome** **Assessment**	**Limitations**	**Main Findings**
Daraki et al., 2018 [22] Greece	Cohort	ADHD symptoms and behavioral difficulties	849/487	4 years	Maternal sera at first or early second trimester	Tertiles	SDQ and Attention-Deficit/Hyperactivity Disorder Test	Outcomes based on parent report only, young age for the assessment of ADHD	Children of mothers in the highest tertile had less hyperactivity–impulsivity symptoms IRR 0.63 (95% CI 0.39–0.99), fewer total ADHD-like symptoms IRR 0.60 (95% CI 0.37–0.95) and a significant score reduction in total behavioral difficulties (beta-coefficient: −1.25, 95% CI −2.32, −0.19) and externalizing symptoms (beta-coefficient: −0.87, 95% CI −1.58, −0.15), compared to the lowest tertile.
Ma et al., 2021 [19] China	Cohort	ADHD symptoms in combination with maternal depression	2552/1125 assessed for ADHD symptoms/145 mothers with depression	4–4.5 years	Umbilical cord blood samples	Continuous and categories: <25 nmol/L and ≥25 nmol/L	Parent version of Conners’ Hyperactivity Index	Age at assessment, subgroup analyses for maternal depression low-powered	No association between low neonatal vitamin D and ADHD symptoms overall, but ADHD symptoms were more likely among vitamin D deficient neonates if the mother had depression (adjusted RR 3.74 (95% CI: 1.49–9.41), compared to the group with no maternal depression.
Morales et al., 2015 [28] Spain	Cohort	ADHD symptoms	3174/1650	Mean age 4.8 years (SD not reported)	Maternal sera, mean gestation weeks of blood draw 13.3 (SD 2.9).	Continuous and categories: <20 ng/mL, 20 to 29.9 ng/mL and ≥30 ng/mL.	Teacher report of ADHD symptoms (using forms with *DSM* and *ICD* criteria)	Up to 40% attrition, symptoms reported only by teachers, age at assessment	Continuous rising vitamin D reduced the number of total ADHD-like symptoms: IRR 0.89 (95% CI 0.80–0.98), inattention subscales: IRR 0.89 (95% CI 0.79–0.99) and hyperactivity–impulsivity scale: IRR 0.88 (95% CI 0.78–0.99).
Mossin et al., 2017 [29] Denmark	Cohort	ADHD symptoms	6707/1233	2–4 years; mean 2.7 years (SD 0.6)	Umbilical cord blood samples	Continuous and categories: <50 nmol/L insufficiency, <25 nmol/L deficiency	Parent-reported CBCL	Age at assessment, only parent report of ADHD symptoms	Vitamin D levels >25 nmol/L and >30 nmol/L were associated with lower ADHD scores compared to levels ≤25 nmol/L (*p* = 0.035) and ≤30 nmol/L (*p* = 0.043), respectively. The adjusted beta coefficient of scoring above the 90th percentile was 0.989 (95% CI 0.979–0.999).
López-Vicente et al., 2019 [37] Spain	Cohort	ADHD symptoms, ASD symptoms, social competence and behavioral problems	3126/2107	5, 8, 14 and 18 years	Maternal sera at mean 13.3 (SD 2.8) weeks of gestation	Continuous and categories: <20 ng/mL, 20 to 29.9 ng/mL and ≥30 ng/mL.	Parent-reported Conner’s Parent Rating Scale, Childhood Autism Spectrum Test, SDQ and CBCL. Teacher-reported California Preschool Social Competence Scale.	High attrition at 14- and 18-years assessments, large number of outcomes but no adjustments for multiple testing	Association between prenatal vitamin D and social competence at 5 years of age: adjusted beta coefficient for 10 ng/mL increment 0.77 (95% CI 0.19–1.35). Null associations with total behavioral problems, ADHD and ASD symptoms in children from 5 to 18 years old.
Vinkhuyzen et al., 2018 [35] The Netherlands	Cohort	Autistic traits	9778/4229 with SRS scores/2489 with both vitamin D measures	~6 years	Maternal sera in mid-gestation and cord blood at birth	Continuous and categories: <25 nmol/L, 25–49.9 nmol/L and ≥50 nmol/L	Parent-reported SRS. Polygenic risk scores accounted for in a subsample.	Differential attrition	Significant associations between vitamin D deficiency and higher scores on the SRS. Mid-gestation vitamin D and SRS scores deficient vs. sufficient: beta coefficient 0.06 (SE 0.01), cord blood: deficient vs. sufficient 0.03 (SE 0.01).
Whitehouse et al., 2013 [18] Australia	Cohort	Autistic traits (parent-reported ASD diagnosis N = 3, not included)	2900/406	Mean age 19.8 years (SD 0.77)	Maternal sera at 18 weeks of gestation	Continuous and categories: ≤49, 50–66, and ≥67 nmol/L	Self-reported autism spectrum quotient questionnaire	High attrition rate, vitamin D categorization (possibly limited power to detect effects of deficiency)	Maternal vitamin D concentrations did not significantly correlate with offspring total autistic traits, nor most questionnaire subscales. There was a weak, inverse correlation with high scores on the attention-switching subscale: OR 5.46 (95 % CI 1.29–23.05) for low vitamin D.
Sullivan et al., 2013 [33] United Kingdom	Cohort	Psychotic symptoms (psychotic disorder outcome excluded)	2399/2047 interviewed/177 with psychotic experiences	18 years	Maternal sera collected at any stage of pregnancy	Continuous and quartiles	The Psychosis-Like Symptom interview	Few mothers were vitamin D deficient (4%).	No association between prenatal vitamin D deficiency and suspected or definite psychotic experiences: OR for fourth quartile 1.00 (95% C1 0.63–1.59) compared to first quartile. No association in continuous analysis either.
Wang et al., 2020 [36] United Kingdom	Cohort	Depressive symptoms	14541/2938 respondents in childhood/2485 in adolescence	Childhood: mean age 10.6 (SD 0.25) and adolescence: mean age 13.8 (SD 0.21)	Maternal sera collected at any stage of pregnancy (median 29.4 weeks)	Categories: <20 ng/mL, 20–29.9 ng/mL and ≥30 ng/mL	Short Moods and Feelings Questionnaire obtained by interview. Polygenic risk scores accounted for in a subsample.	Differential attrition	No association between prenatal vitamin D deficiency and depressive symptoms in childhood: OR 1.07 (95% CI 0.73–1.58) nor adolescence: OR 1.32 (95% CI 0.98–1.79). No interactions with polygenic risk scores.
Chawla et al., 2019 [39] USA	Cohort	Internalizing and externalizing symptoms, dysregulation, ASD symptoms	1700/218	1–2 years; mean 14.3 months (SD 3.3)	Maternal sera in first or second trimester	Quartiles	Parent-reported Infant Toddler Social Emotional Assessment	Results reported only for different ethnic groups, not for the overall sample. Multiple testing not accounted for. Low power in ethnic subgroups.	Mixed findings. Lower prenatal vitamin D was associated with higher (less favorable) internalizing scores among white infants, but the opposite among Black and Hispanic infants. Among Black infants only, lower vitamin D was associated with higher (more favorable) ASD Social Competence scores and lower (more favorable) ASD problem scores. No strong patterns for externalizing symptom scores.
Whitehouse et al., 2012 [17] Australia	Cohort	Internalizing and externalizing behavior	2900/743	2, 5, 8, 10, 14 and 17 years	Maternal sera at 18 weeks of gestation	Quartiles	Parent-reported CBCL	Attrition in the adolescent samples. Exclusion of non-participants	There were no significant associations between prenatal vitamin D concentrations and offspring behavioral or emotional difficulties at any time point in childhood or adolescence (no ORs provided, all p values non-significant).
Allen et al., 2013 [16] Australia	Cohort	Eating disorder symptoms	2900/526 respondents of which 98 had eating disorder symptoms	14–20 years	Maternal sera at 18 weeks’ gestation	Quartiles	Child Eating Disorder Examination and Eating Disorder Examination-Questionnaire	Small sample size	Low maternal vitamin D levels were associated with offspring eating disorder symptoms in females only aOR 2.09 (95% CI: 1.03–5.27). No association in the overall sample.

Abbreviations: aOR: adjusted odds ratio, CBCL: Child Behaviour Checklist, CI: confidence interval, DSM: Diagnostic and Statistical Manual of Mental Disorders, ICD: International Classification of Diseases, IRR: incidence rate ratio, OR: odds ratio, RR: relative risk, SD: standard deviation, SDQ: Strengths and Difficulties questionnaire, SE: standard error, SRS: Social Responsiveness Scale.

**Table 2 ijms-24-00063-t002:** Findings of the reviewed studies by each outcome.

Outcome	Positive Findings	Mixed Findings	Null Findings
ADHD	Sucksdorff et al., 2021 [32] Chu et al., 2022 [44]	0	Chu et al., 2022 [44] Gustafsson et al., 2015 [26] Strøm et al., 2014 [31]
ADHD symptoms	Daraki et al., 2018 [22] Morales et al., 2015 [28] Mossin et al., 2017 [29]	0	López-Vicente et al., 2019 [37] Ma et al., 2021 [19]
ASD	Chen et al., 2016 [20] Fernell et al., 2015 [25] Lee et al., 2021 [38] Sourander et al., 2021 [30] Vinkhuyzen et al., 2017 [34] Wu et al., 2018 [21]	Lee et al., 2021 [38] Schmidt et al., 2019 [41] Windham et al., 2019 [42] Windham et al., 2020 [43]	Egorova et al., 2020 [23] Vinkhuyzen et al., 2017 [34]
ASD symptoms	Vinkhuyzen et al., 2018 [35] Vinkhuyzen et al., 2018 [35]	Chawla et al., 2019 [39]	López-Vicente et al., 2019 [37] Whitehouse et al., 2013 [18]
Depressive disorder	0	0	Strøm et al., 2014 [31]
Depressive symptoms	0	0	Wang et al., 2020 [36]
Eating disorder symptoms	0	Allen et al., 2013 [16]	0
Schizophrenia or schizoaffective disorder	Eyles et al., 2018 [24] McGrath et al., 2010 [27]	0	McGrath et al., 2003 [40]
Psychotic experiences	0	0	Sullivan et al., 2013 [33]
Behavioral or emotional symptoms	Daraki et al., 2018 [22]	Chawla et al., 2019 [39]	López-Vicente et al., 2019 [37] Whitehouse et al., 2012 [17]

**Table 3 ijms-24-00063-t003:** Large studies with stricter definitions for vitamin D deficiency. Only studies that had defined vitamin D deficiency as ≤40 nmol/L and had at least 100 mothers in the deficiency group were included. An additional limit for the diagnostic outcomes was to have at least 100 cases for the studied outcome.

Outcome	Positive Findings	Null Findings
ADHD	Sucksdorff et al., 2021 [32]	0
ADHD symptoms	Daraki et al., 2018 [22] Morales et al., 2015 [28] Mossin et al., 2017 [29]	López-Vicente et al., 2019 [37] Ma et al., 2021 [19]
ASD	Lee et al., 2021 [38] Sourander et al., 2021 [30] Wu et al., 2018 [21]	0
ASD symptoms	Vinkhuyzen et al., 2018 [35] Vinkhuyzen et al., 2018 [35]	López-Vicente et al., 2019 [37]
Depressive disorder	0	0
Depressive symptoms	0	0
Eating disorder symptoms	0	0
Schizophrenia or schizoaffective disorder	Eyles et al., 2018 [24]	0
Psychotic experiences	0	0
Behavioral or emotional symptoms	Daraki et al., 2018 [22]	López-Vicente et al., 2019 [37]

## Data Availability

Not applicable.

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
