# Peer review of "Maternal Vitamin D Levels during Pregnancy and Offspring Psychiatric Outcomes: A Systematic Review"

_ijms, 2022, doi:10.3390/ijms24010063_

Round 1
Reviewer 1 Report
This is a systematic review of maternal vitamin D levels during pregnancy or at birth (cord blood) and offspring neuropsychiatric and psychiatric outcomes. Specific possible outcomes identified for the study were autism-spectrum disorders, ADHD, schizophrenia, depression and eating disorders. It is true that nearly every cell in the human body has vitamin D receptors but the relationship between brain development/pathology and vitamin D deficiency has only been demonstrated in animal models. There are no randomized controlled studies of the impact of perinatal vitamin D levels with maternal vitamin D supplements and neuropsychiatric outcomes-- all the studies reviewed in this paper are observational in nature. As the authors themselves point out this is problematic given the very heterogenous nature of the studies (see lines 174-187), especially the variability of the timing in pregnancy of the vitamin D levels, and the significant concerns about the very definition of vitamin D deficiency, vitamin insufficiency (lower vit D levels) using 25-OH vitamin D levels (see Cummings SR, Rosen C. NEJM 2022;387:368-370). There is also the likelihood of other confounding variables including the potential for other maternal micronutrient deficiencies, genetic differences (family history of psychopathology), and socio-economic conditions. Thus the overall quality of the evidence supporting the limited observations of this study on the association of schizophrenia (4 studies) and autism spectral disorders (ten studies), with lower maternal vitamin D levels or even “deficiency”, is low. As it turns out very few mothers in the database had serum 25-OH vitamin D levels less than 20 nmol/L. Though there has been an explosive number of publications on the impact of vitamin D “deficiency or insufficiency” and human health, the only convincing impacts are still limited to those affecting bone health.
Other comments:
1. Figure 1—a footnote is needed here to explain “28+1”.
2. Figure 2—given the limited, 20 year time frame, this figure adds nothing to this paper.
3. Methods—it is not clear why various “symptoms” were used as outcome measures—presumably these were from parental reports or questionnaires, and as they resulted in no significant findings/associations with vitamin D levels, why are they even included in this study? As it appears that only a single study reported on depression or depressive symptoms, why is this outcome included at all?
4. Lines 423-426. Please consider a rewrite of these sentences: Our systematic review provides some evidence for the association between lower maternal vitamin D levels during pregnancy and offspring ASD and schizophrenia. If confirmed, the findings could have implications for public health, as maternal vitamin D levels are very responsive to vitamin D supplements.
5. I would suggest that if the authors concluded that there was" little" rather than "some" support of the impact of vitamin D it would make ore of a contribution to the literature. t
Reviewer 2 Report
This is a very interesting study that addresses the potential use of vitamin D as an indicator of future mental illness at birth.
I have some doubts and comments about the authors.
Lines 27-33: Vitamin D is repeated too many times in the paragraph which makes it difficult to read.
The search includes articles up to August 10, 2022. But the start date is 2003. Why this date?
In the design, the number of studies includes, why 28+1?
Line 403: A is repeated
I do not agree with the sentence " Our systematic review provides some evidence for the association between maternal vitamin D levels during pregnancy and offspring ASD, possibly also for ADHD and schizophrenia."
I would remove that categorical conclusion, What do the authors mean by "some evidence"?
I don't think you can conclude by mixing the ages of offspring ranging from infancy to adulthood. Many other factors may have played a role.
Is there any idea of the vitamin D level at birth, and has any correlation been seen between that of the mother and the "possible association" with mental disease development?
Round 2
Reviewer 1 Report
I appreciate the authors revisions. I have two more suggestions:
1. First line of abstract should read something like this: Prenatal exposure to vitamin D may play a significant role in human brain development and function.
2. The first line of the introduction should read something like this: Vitamin D is essential for bone mineralization and bone mass acquisition. It may also be important for the development of other organ systems including the central nervous system/brain.
Author Response
Reviewer 1 comments:
I appreciate the authors revisions. I have two more suggestions:
- First line of abstract should read something like this: Prenatal exposure to vitamin D may play a significant role in human brain development and function.
Response: We updated the first line of abstract as suggested.
- The first line of the introduction should read something like this: Vitamin D is essential for bone mineralization and bone mass acquisition. It may also be important for the development of other organ systems including the central nervous system/brain.
Response: We updated the first line of introduction as suggested.
Reviewer 2 Report
After reading the authors' responses, I still think that this systematic review is not sufficiently supported, neither in the search inclusion criteria nor in the conclusions.
In my opinion, there are too many confounding variables, including the possibility of other maternal micronutrient deficiencies, genetic differences (family history of psychopathology) and socioeconomic conditions.
Author Response
Reviewer 2 comments:
After reading the authors' responses, I still think that this systematic review is not sufficiently supported, neither in the search inclusion criteria nor in the conclusions.
In my opinion, there are too many confounding variables, including the possibility of other maternal micronutrient deficiencies, genetic differences (family history of psychopathology) and socioeconomic conditions.
Response: We agree that neuropsychiatric and psychiatric disorders have complex etiology, which is impacted by various factors. By this systematic review, we do not aim to prove any causality but merely gather all the literature there is so far regarding the associations found between perinatal vitamin D levels and (neuro)psychiatric outcomes. These reviewed studies had considered different sets of confounders, for example sociodemographic factors, perinatal factors and parental psychopathology. We have reported the limitations of the included confounders in these studies in the results and now sentences have also been added in the discussion. We understand that many other factors may have their role as confounders, antecedents or mediators, but these cannot be named or examined in this review, but to be further studied in the future.
Covariates have been previously reported in results, page 5, lines 143-146.
“The covariates included in the studies varied greatly. Most studies included prenatal, perinatal and socio-demographic factors, but maternal psychopathology was addressed in 10 studies [19,21,24,27,29,30,32,33,36,38].”
Added these in the discussion: page 6, line 386-388.
“The included confounders varied: most studies had considered socio-demographic and pre- or perinatal risk factors, but parental psychopathology was included only in ten studies. Parental psychopathology is an important potential confounder because genetic effects are well-established in the etiology of most neuropsychiatric and psychiatric disorders.”
Added following sentence in conclusions: Page 8. line 467-468.
“Several other factors including genetic factors also play a role in the multifactorial etiology of neuropsychiatric and psychiatric disorders.”

Round 3
Reviewer 2 Report
If the editor considers this manuscript suitable for publication, I have no objection.